# Late gene therapy limits the restoration of retinal function in a mouse model of retinitis pigmentosa

Miranda L. Scalabrino [1,2,4], Mishek Thapa[1,2,4], Tian Wang[3], Alapakkam P. Sampath [1], Jeannie Chen[3] & Greg D. Field [1,2] ✉

Retinitis pigmentosa is an inherited photoreceptor degeneration that begins with rod loss followed by cone loss. This cell loss greatly diminishes vision, with most patients becoming legally blind. Gene therapies are being developed, but it is unknown how retinal function depends on the time of intervention. To uncover this dependence, we utilize a mouse model of retinitis pigmentosa capable of artificial genetic rescue. This model enables a benchmark of best-case gene therapy by removing variables that complicate answering this question. Complete genetic rescue was performed at 25%, 50%, and 70% rod loss (early, mid and late, respectively). Early and mid treatment restore retinal output to near wild-type levels. Late treatment retinas exhibit continued, albeit slowed, loss of sensitivity and signal fidelity among retinal ganglion cells, as well as persistent gliosis. We conclude that gene replacement therapies delivered after 50% rod loss are unlikely to restore visual function to normal. This is critical information for administering gene therapies to rescue vision.

Current gene therapies for photoreceptor degeneration can slow disease progression, but thus far, nothing fully stops cell death in preclinical models or patients[1,2]. One potential reason is that gene therapy involves several challenging technical confounds including viral design, mode of delivery, appropriate expression of the transgene, and stochasticity in cellular infection, all of which can produce abnormal levels of the therapeutic gene and/or limit the population of treated cells. By optimizing these parameters, it is possible to achieve complete gene delivery in surviving cells and normal levels of gene expression, which could fully stop photoreceptor death and restore normal vision. Alternatively, it is possible that once some amount of photoreceptor death occurs, gene therapy has a limited effect on halting additional cell loss, or on rescuing visual function.

To examine this issue, we used a mouse model of gene therapy for retinitis pigmentosa (RP) in which complete genetic rescue across the retina can be achieved without the use of viral vectors. This mouse

does not express the beta subunit of the rod-photoreceptor cGMP-gated (CNG) channel because a neomycin cassette flanked by *loxP* sites was inserted into intron 19 of the *Cngb1* locus (*Cngb1^neo/neo*)[3–5]. This prevents CNGB1 expression and reduces the formation of CNG channels, ultimately leading to rod degeneration and death similar to humans with RP[6]. All rods are lost by ~6 months postnatal; cones begin to die at 3–4 months and are all lost by ~8 months[4,7]. To mimic gene therapy, we crossed this *Cngb1^neo/neo* line with the *UBC-cre-Ert* line containing tamoxifen-inducible cre recombinase[8]. By delivering tamoxifen, activated cre removes the neomycin insert to enable endogenous CNGB1 expression across all rods, thereby producing a best-case scenario for gene therapy[9].

We used this model to determine the extent to which photoreceptor survival and retinal signaling depended on the level of rod death prior to rescuing CNGB1 expression. We induced genetic rescue at multiple ages corresponding to different amounts of rod and cone

[1]Stein Eye Institute, Department of Ophthalmology, University of California, Los Angeles, CA, USA. [2]Department of Neurobiology, Duke University School of Medicine, Durham, NC, USA. [3]Zilkha Neurogenetic Institute, Keck School of Medicine, University of Southern California, Los Angeles, CA, USA. [4]These authors contributed equally: Miranda L. Scalabrino, Mishek Thapa. ✉e-mail: gregfield@ucla.edu

photoreceptor loss in animals of both sexes. First, we assayed retinal structure across three treatment timepoints and 4 posttreatment timepoints (12 cohorts), as well as controls from age-matched untreated or heterozygous littermates (48 mice total). We treated mice at 1, 2, and 3 months of age, corresponding to ~25%, 50%, and 70% rod loss, and 0%, 0%, and 5% cone loss, respectively. Following treatment, we measured the amount of additional photoreceptor loss and the presence of retinal inflammation. We also measured changes in the fidelity of retinal signaling and receptive field structure of retinal output neurons, the retinal ganglion cells (RGCs), using a large-scale multielectrode array. Importantly, the structural data and physiology data were collected from the same retinas, allowing a comparison between structural and functional recovery following treatment.

Following each treatment timepoint, there was some persistent, but modest photoreceptor loss several months after treatment. However, late treatment exhibited persistent gliosis, indicative of inflammation, while this was not observed following early and mid-treatment. Functionally, there was a striking difference between the early (1 M) and mid (2 M) treatments versus late (3 M) treatment in terms of RGC function at cone-mediated light levels. The gain and fidelity of RGC signals recovered to levels nearly indistinguishable from wild-type (WT) following early and mid-treatments, but they failed to recover following late treatment. Similar results were observed at rod-mediated light levels, with early and mid-treatment resulting in nearly normal RGC signaling while late treatment exhibited continued deterioration. These results indicate that even under best-case scenarios, gene therapies for photoreceptor degeneration ought to be delivered prior to ~70% rod loss for long-term vision restoration and that gene therapy for RP may not perform well under conditions with more than 50% rod loss. Thus, the timing of genetic rescue is a critical variable for restoring vision.

## Results

### Genetic rescue slows but does not immediately stop photoreceptor death

To assess the extent to which the timing of therapy impacts rod-photoreceptor rescue, we induced genetic rescue at three timepoints using tamoxifen chow. Early treatment was performed in mice 1 month (1 M) of age with ~25% rod loss (Fig. 1A) and no cone loss (Fig. 1C). The mid-treatment consisted of 2 M mice with ~50% rod loss and no cone loss, and the late treatment was performed in 3 M mice with ~70% rod loss and ~5% cone loss (Fig. 1A, C) (prior analysis of cre-mediated recombination provided in ref. 9 and prior quantification of cone loss provided in ref. 4). Treated mice were sacrificed at 1-month intervals between ages 4 M and 7 M to measure the visual response properties of RGCs and to histologically assess the time-dependent effects of treatment on photoreceptor survival. We have shown that 7-day tamoxifen treatment produces robust cre-mediated recombination across surviving rods[9], and treatment restored CNGA1 expression, indicating genetic rescue resulted in intact channel formation (Supplementary Fig. S1). At all timepoints, tamoxifen treatment reduced rod and cone loss compared to untreated animals (Fig. 1), with greater preservation in dorsal retina (Supplementary Fig. S2). However, there was a weak trend showing continued photoreceptor loss several months following treatment, as measured by the number of nuclei in the outer nuclear layer (Fig. 1E). With early treatment, $76 \pm 8\%$ of photoreceptors remain at 4 M, but this dropped to $65 \pm 3\%$ by 7 M ($P$ value = 0.007, 1624 counted nuclei and 48 sampled regions). Next, with mid-treatment, $51 \pm 7\%$ of photoreceptors remain at 4 M and decreased to $41 \pm 1\%$ by 7 M ($P$ value: 0.032, 1164 counted nuclei and 48 sampled regions). Finally, with late treatment, $30 \pm 6\%$ of photoreceptors remain at 4 M and decreased to $26 \pm 1\%$ at 7 M ($P$ value: 0.106, 739 counted nuclei and 51 sampled regions). These results suggest photoreceptor health is not fully stabilized following a treatment that cures the initial cause of rod death, though degeneration is

substantially slowed. Surprisingly, the initial rate of continued cell death may be faster with the earlier treatment (Fig. 1E, early). Despite these lower numbers of photoreceptors following treatment, outer segments of remaining rods (Fig. 1B) and cones (Fig. 1D) persisted.

### Synaptic structure is disorganized in late therapy

Rod death causes many secondary changes, including retinal rewiring, particularly in the outer plexiform layer[10–13]. Thus, we assessed how the structure of synaptic terminals depended on the treatment timepoint. Early treatment largely preserved rod and cone terminals, as assessed by staining for CtBP2, which is the major structural component of the synaptic ribbon between photoreceptors and bipolar cells (Fig. 2A, B). CtBP2 labeling indicated normal horseshoe-like synaptic structures were prevalent in retinas receiving the early treatment[14,15]. For mid-treatment, CtBP2 labeling continued to reveal horseshoe structures. Following late treatment, CtBP2 horseshoe structures were substantially reduced despite ~25% of rods and ~90–95% of cones remaining at this treatment timepoint. The sparser synapses following late treatment suggest that earlier treatment may be important for rescuing normal and long-lasting retinal function (see "Discussion").

### Gliosis present in late-treated retinas

As a final measure of retinal structure following the time-dependent rescue of *Cngb1*, we investigated the activation of Müller glia by immunolabeling glial fibrillary acid protein (GFAP)[16]. This marker indicates a retinal inflammatory response, which is present in patients with RP[17–20]. While this response can protect the retina from damage (e.g., by releasing neuroprotective molecules)[21,22], prolonged activation can increase tissue damage through the release of pro-inflammatory markers[23]. We found GFAP labeling was minimally present in early or mid-treatment retinas examined at 7 M, as well as in tamoxifen-treated WT retinas. However, we found that GFAP was present in both untreated and late-treated retinas at 7 M (Fig. 3). This indicates a prolonged retinal stress response and pathology despite genetic rescue: GFAP labeling was present 4 months after the late-treatment timepoint. Importantly, this is independent of viral gene therapy, which is also known to invoke an immune response. Thus, late treatment did not markedly improve glia-induced retinal inflammation.

### Late genetic rescue fails to restore gain of RGC responses at cone-mediated light levels

The preceding histological assessments suggest that retinal function might be compromised with late treatment relative to the early and mid-treatment timepoints. However, previous studies have also indicated that retinal function can remain relatively robust despite marked changes in photoreceptor morphology and density in the *Cngb1*$^{neo/neo}$ and *rd10* models of RP[4,5]. To determine the impact of treatment timepoint on retinal function, we measured visual responses among RGCs, the output neurons of the retina, using a large-scale multi-electrode array (MEA)[24–26]. In total, we measured responses from 22,783 RGCs. We focused on RGCs because changes in their response properties capture the net effects of degeneration and treatment on the signals transmitted from the retina to other brain areas[4]. We began by comparing the receptive field properties of RGCs following the early, mid, and late-treatment timepoints. Receptive fields summarize the spatial and temporal integration performed by RGCs and their presynaptic circuits as well as summarize the visual features that are signaled by RGCs to the brain[27,28]. We measured receptive fields at a photopic (cone-mediated) light level (10,000 Rh*/rod/s). We also focused on measuring the impact of rod *Cngb1* rescue on cone-mediated vision because we have shown previously that *Cngb1* rescue restores normal rod signaling[9]. Furthermore, the impact on cone-mediated vision is likely to be most informative and impactful for human-directed gene therapies.

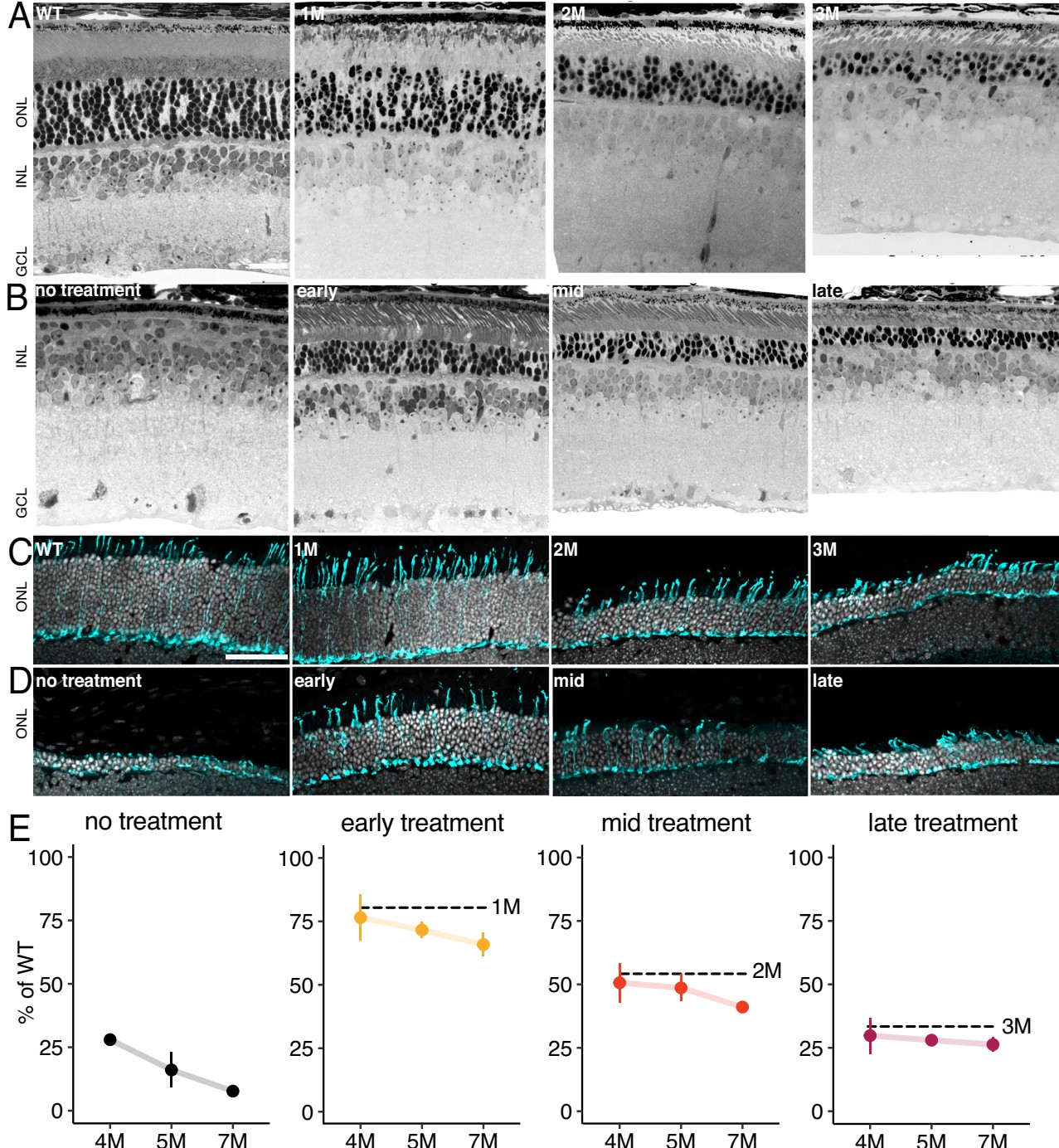

**Fig. 1 | Photoreceptor degeneration continues despite genetic rescue. A** Light microscope images of retinal cross-sections from wild-type (WT) and *Cngb1^neo/neo* mice at each treatment timepoint. Scale bar: 20 μm. **B** Cross-sections from an untreated (*Cngb1^neo/neo*;cre-) retina (left) at 5 M and three treated retinas after early, mid, or late treatment, each at 5 M. **C** Confocal images of retinal cryosections from WT and *Cngb1^neo/neo* mice at each treatment timepoint. Cone structure in cyan (mCar) and nuclei in white (DAPI). Scale bar: 50 μm. **D** Cross-sections from an untreated (*Cngb1^neo/neo*;cre-) retina (left) at 7 M and three treated retinas after early, mid, or late treatment, each at 7 M. **E** Quantification of outer nuclear layer (ONL) cell counts over time to measure fraction of surviving photoreceptors. Dashed line indicates ONL thickness at time of treatment. Error bars and points indicate mean +/−1 standard error. Measurements are from 5074 nuclei across n = 210 sampled regions (each region 1000 μm²). Retinas were sampled repeatedly (5–8 retinas per treatment group and 1–3 retinas per timepoint of assay). Source data are provided as a Source Data file.

To measure the receptive fields of RGCs, we presented checkerboard noise stimuli while recording RGC spikes with a large-scale MEA. In a typical experiment, we measured the responses of 250–430 RGCs simultaneously. Spatiotemporal receptive fields were estimated by computing the spike-triggered average to the checkerboard stimulus[27]. This provides an estimate of the linear component of the spatiotemporal receptive field. To analyze separately, the changes in spatial and temporal receptive fields, we focused our analysis on RGCs with space-time separable spike-triggered averages (see "Methods")[4]. Variability in receptive field measurements may arise from variability in experimental preparations, sex of animals, and neuronal cell type. As such, a mixed-effects model was used to

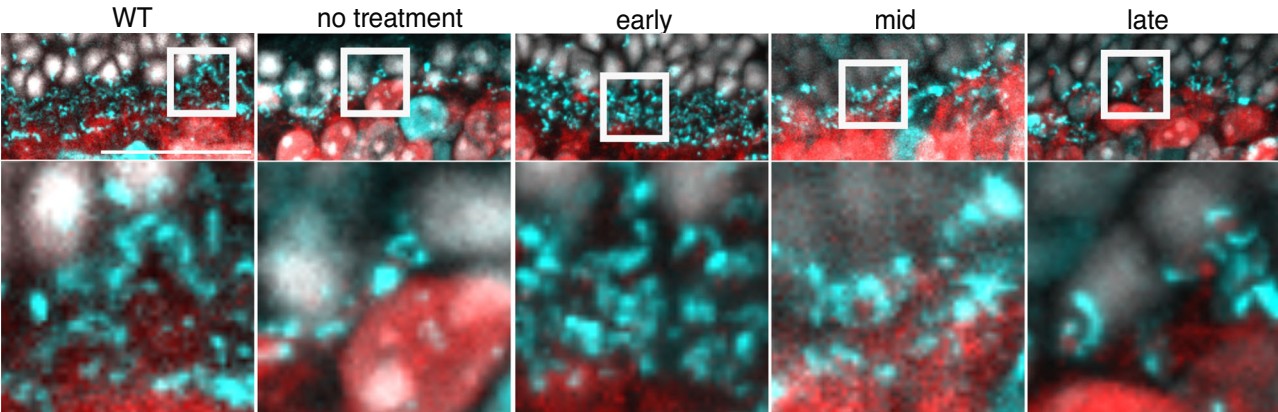

**Fig. 2 | Photoreceptor synapses are reduced but stable following treatment.** Histology of rod and cone synapses in cyan (CtBP2) at 7 M from (from left to right) WT, untreated (*Cngb1^neo/neo*;cre-), early, mid and late-treatment timepoints. Nuclei in white (DAPI) and rod ON bipolar cells in red (PCP2) to visualize approximate location of bipolar cell bodies and dendrites. The square region in top row enlarged below. Scale bar: 50 μm. Representative images chosen from two to four biological replicates imaged per cohort.

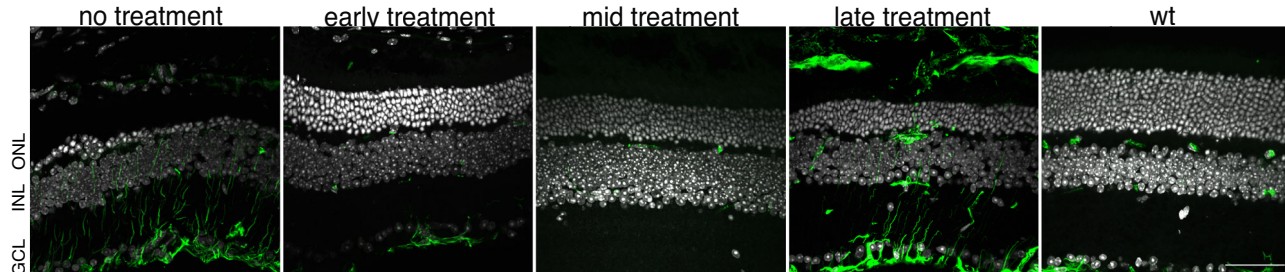

**Fig. 3 | Activated glia present in late-treated retina.** Immunofluorescence shows GFAP (green) expression in 7 M retinas (left to right): WT, untreated (cre-), early, mid, and late-treatment timepoints. Nuclei labeling with DAPI shown in white. Scale bar: 50 μm. ONL outer nuclear layer, INL inner nuclear layer, GCL ganglion cell layer. Representative images chosen from two to four biological replicates imaged per cohort.

determine associations between treatment timepoint and RGC physiology to account for potential confounds (see "Methods"). Changes in photopic temporal receptive fields were subtle in all treated retinas, largely because the temporal receptive fields changed little between 4 and 7 M in untreated animals under photopic conditions (Table 1, line 1) (Fig. 4A)[4]. The biggest change was in the late-treatment group where receptive fields appeared to slow relative to WT at the 5 M timepoint, but even this change was small and was not statistically significant (Table 1, line 2) (Fig. 4A). Thus, the duration of temporal integration within RGC receptive fields was relatively stable following genetic rescue and did not depend on the treatment timepoint between 1 M and 3 M (25–70% rod loss).

We also analyzed the size of spatial receptive field centers across RGCs. Like the temporal receptive fields, spatial receptive fields from treated animals were similar in size to WT (Table 1, line 3) and did not decrease over time (for example, at early rescue, 2% difference in mean was observed between 4 and 7 M; Table 1, line 4) (Fig. 4B); this is likely because there were minimal changes to the spatial receptive fields of untreated animals at even at 7 M under photopic conditions (Fig. 4B, gray distributions). These results indicate that spatial and temporal receptive field structure between 4 and 7 M post treatment do not depend strongly on the treatment timepoint. This is not particularly surprising because cone-mediated receptive field structure is relatively robust even to severe rod loss and changes in cone morphology[4,5].

We next examined the contrast response functions of the RGCs. Also referred to as static nonlinearities in reverse correlation analyses[27], the contrast response functions capture how many spikes an RGC tends to produce for a given similarity (correlation) between the stimulus and the receptive field. In untreated retinas, there was a diminished gain between 4 and 7 M relative to WT retinas (Fig. 4C, gray distributions). Thus, genetic rescue had the potential to improve response gain and transform visual responses to be more similar with WT. Indeed, early treatment improved gain to near WT levels by 7 M (14% difference; Table 1, line 5) (Fig. 4C). Mid-treatment also improved gain to near WT levels at 7 M (15% difference; Table 1, line 6). Interestingly, for both early and mid-treatment, gain rose over several months following treatment, suggesting this increase resulted from both changes in photoreceptor health and retinal wiring (see "Discussion").

Unlike early and mid-treatment, late treatment produced a modest recovery of gain that was higher than untreated animals at 7 M (at 7 M, 20% higher relative to 4 M; Table 1, line 7), but substantially lower than animals treated at 1 M or 2 M (Fig. 4C, Table 1, lines 8–9). Qualitatively similar results were obtained at a mesopic light level (100 Rh*/rod/s) (Supplementary Fig. S3). Thus, this late-treatment timepoint was ineffective at restoring the contrast response gain at cone-mediated light levels despite 25% of the rods and 95% of the cones remaining at the time of treatment[4].

## Late genetic rescue results in higher noise

Above, we showed that late treatment results in reduced response gain across the RGC population. We wondered if there was also a change in RGC response variability, or noise. This is important because greater noise in the response will result in less reliable signaling of visual information. There are two potential sources of increased noise: signal-independent and signal-dependent. Signal-independent noise manifests as increased spontaneous activity in RP[29–31]. However, we previously found that *Cngb1^neo/neo* mice do not exhibit increases in

**Table 1 | Statistics for comparisons between experimental groups**

| Line | Group 1 | Group 1 mean +/− SEM | Group 2 | Group 2 mean +/− SEM | P value |
|---|---|---|---|---|---|
| 1 | Untreated 4 M Cngb1neo/neo | 0.19+/−0.003 ms | Untreated 7 M Cngb1neo/neo | 0.17+/−0.003 ms | 0.12 |
| 2 | Late-treatment 5 M Cngb1neo/neo | 0.20+/−0.003 ms | WT | 0.22+/−0.002 ms | 0.13 |
| 3 | Late-treatment 7 M Cngb1neo/neo | 7984+/−162 µm² | WT | 8415+/−196 µm² | 0.44 |
| 4 | Early-treatment 4 M Cngb1neo/neo | 10739+/−148 µm² | Early-treatment 7 M Cngb1neo/neo | 10315+/−221 µm² | 0.34 |
| 5 | Early-treatment 7 M Cngb1neo/neo | 33.31+/−0.71 spk/s | WT | 36.6+/−0.67 spk/s | 0.11 |
| 6 | Mid-treatment 7 M Cngb1neo/neo | 36.01+/−0.71 spk/s | WT | 36.6+/−0.68 spk/s | 0.24 |
| 7 | Late-treatment 4 M Cngb1neo/neo | 21.4+/−0.77 spk/s | Late-treatment 7 M Cngb1neo/neo | 24.4+/−0.81 spk/s | 0.01 |
| 8 | Late-treatment 7 M Cngb1neo/neo | 24.4+/−0.81 spk/s | Mid-treatment 7 M Cngb1neo/neo | 36.0+/−0.71 spk/s | <0.001 |
| 9 | Late-treatment 7 M Cngb1neo/neo | 24.4+/−0.81 spk/s | Early-treatment 7 M Cngb1neo/neo | 33.3+/−0.71 spk/s | <0.001 |
| 10 | Untreated 7 M Cngb1neo/neo | 1.86+/−0.07 spk/s | WT | 0.989+/−0.024 spk/s | <0.001 |
| 11 | Early-treatment 7 M Cngb1neo/neo | 0.968+/−0.033 spk/s | WT | 0.989+/−0.024 spk/s | 0.36 |
| 12 | Mid-treatment 7 M Cngb1neo/neo | 1.01+/−0.024 spk/s | WT | 0.989+/−0.024 spk/s | 0.43 |
| 13 | Untreated 4 M Cngb1neo/neo | 18.59+/−0.35 bits/s | Early-treatment 4 M Cngb1neo/neo | 21.33+/−0.35 bits/s | 0.001 |
| 14 | Untreated 4 M Cngb1neo/neo | 18.59+/−0.35 bits/s | Mid-treatment 4 M Cngb1neo/neo | 20.35+/−0.58 bits/s | 0.01 |
| 15 | Late-treatment 4 M Cngb1neo/neo | 19.83+/−0.27 bits/s | Late-treatment 7 M Cngb1neo/neo | 16.05+/−0.42 bits/s | <0.001 |
| 16 | Early-treatment 7 M Cngb1neo/neo | 18.81+/−0.25 bits/s | WT | 19.42+/−0.25 bits/s | 0.82 |
| 17 | Early-treatment 4 M Cngb1neo/neo | 16.6+/−0.41 bits/s | WT | 19.42+/−0.25 bits/s | 0.02 |
| 18 | Late-treatment 4 M Cngb1neo/neo | 14.67+/−0.41 bits/s | Late-treatment 7 M Cngb1neo/neo | 12.82+/−0.48 bits/s | 0.03 |
| 19 | Late-treatment 7 M Cngb1neo/neo | 12.82+/−0.48 bits/s | WT | 19.42+/−0.25 bits/s | <0.001 |

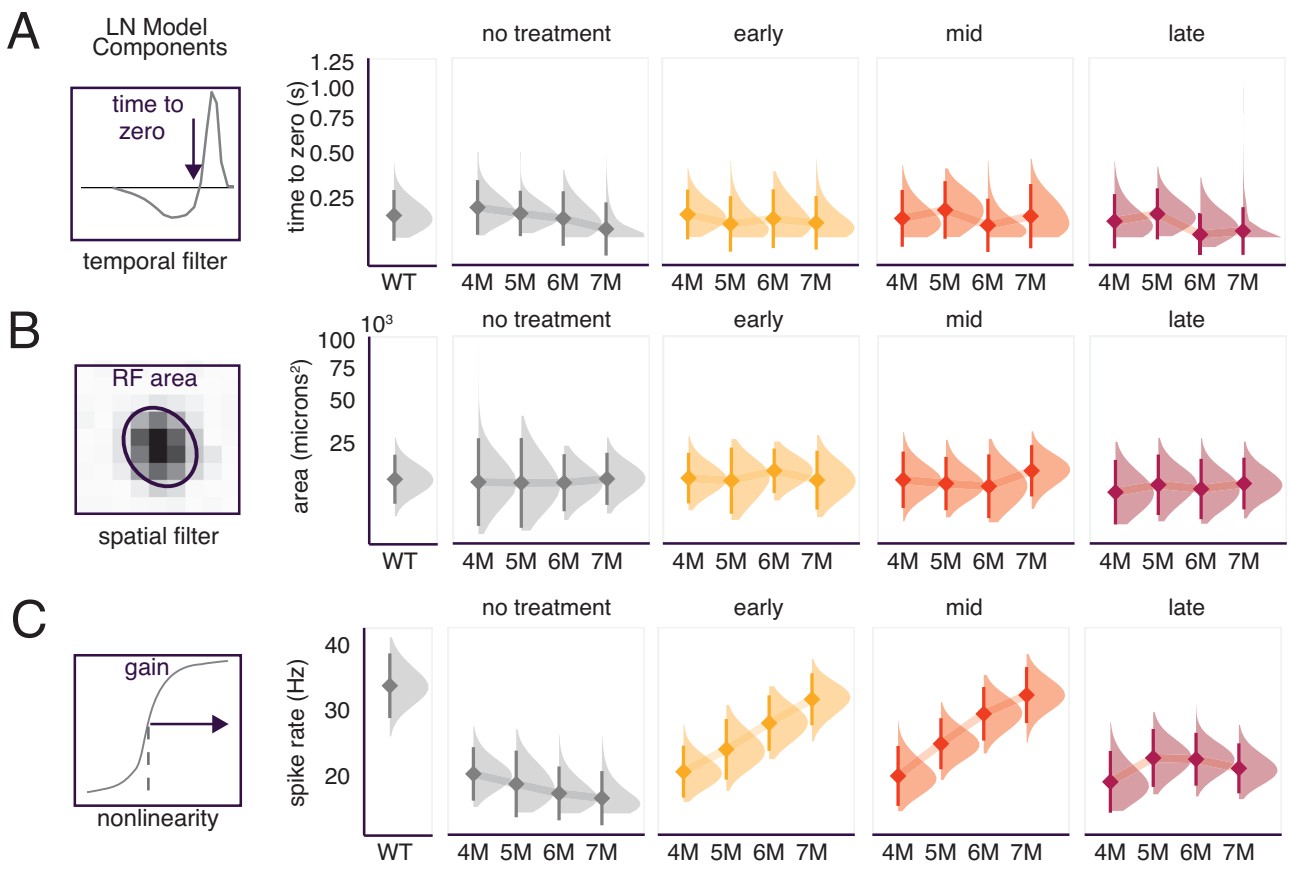

**Fig. 4 | Receptive fields are stable following treatment, but response gain is not recovered following late treatment. A** Left: example temporal receptive field (RF) of an RGC, showing how the stimulus (on average) changed immediately preceding a spike. Arrow indicates the time-to-zero used to estimate the integration time. Right: panels show distributions of the time-to-zero (diamond is the mean and bar is +/− 2 SD) in WT, no treatment, early, mid and late-treatment retinas from 4 to 7 M. **B** Left: Spatial RF of an example RGC, ellipse is the 2-sigma contour of a two-dimensional Gaussian fit. The geometric mean of the major and minor axes of the ellipse was used to estimate RF area, right: same as A-right but for RF size. **C** Left: example contrast response function. Spike rate at 50% of peak was used to estimate gain. Right: same a A-right but for gain estimates. *n* = 48 retinas. Light level 10,000 Rh*/rod/s. Source data are provided as a Source Data file.

spontaneous activity or spontaneous oscillations until nearly all the photoreceptors have died (8–9 M in untreated animals)[4], indicating there is not an increase in signal-independent noise.

To examine signal-dependent noise, we measured the variance in the spike count while presenting a repeated checkerboard stimulus (Fig. 5A). Higher variance in the spike counts to a repeated stimulus indicates an increase in signal-dependent noise given no change in signal-independent noise. For a Poisson process, the variance in spike rate is equal to the mean spike rate (Fig. 5B, C, solid line). WT responses fell along this line (black dots, Fig. 5B–D), indicating they were approximately consistent with a Poisson process when using a checkerboard noise stimulus. However, responses from *Cngb1*[neo/neo] mice exhibited a clear tendency to lie above this line, indicating higher variance responses for a given mean (Fig. 5B, D, Table 1, line 10). Early and mid-treatment brought the response variability back toward that of WT (Table 1, line 11–12). However, late treatment failed to reduce the signal-dependent noise, which lingered near that of untreated *Cngb1*[neo/neo] RGCs even at 7 months post treatment (Fig. 5C, D, F). These results indicate that the treatment timepoint is critical for robustly and stably reducing variability in cone-mediated RGC responses.

### Late treatment fails to rescue visual information

Thus far we have shown that late treatment at 3 M (70% rod loss) fails to restore the gain of RGC responses to WT or near WT levels (Fig. 4C) and results in increased response variability to a checkerboard stimulus (Fig. 5D). Decreased gain and increased variability should result in less information transmission from the retina to the brain. To assess directly the information content of RGC responses, we presented a repeated 10 s checkerboard stimulus (see "Methods") and calculated the mutual information between RGC responses and the stimulus[4,32]. Mutual information indicates how much observing an RGC response

reduces uncertainty about the stimulus[33]. RGCs with highly reproducible responses will generally provide more information about a stimulus (Fig. 6A, RGC 1) than those with less reproducible responses (Fig. 6A, RGC 2). In treated retinas, the cone-mediated information rate from early and mid-treatment retinas increased by 15% and 9.5%, respectively, relative to untreated retinas measured at 4 M (Table 1, lines 13–14), and approached WT levels by 6 M (Fig. 6B). However, in the late-treated retinas, information declined over time (19% decrease in cone-mediated information transmission at 7 M relative 4 M; Table 1, line 15) (Fig. 6B). In sum, late treatment resulted in information rates that were substantially greater than untreated animals, but also far less than earlier treated and WT animals. These results further indicate that late treatment of rod degeneration does not ultimately restore normal cone-mediated signaling among RGCs.

We have shown previously that rescuing CNGB1 expression in rods restores rod light responses and dim-flash sensitivity among RGCs[9]. However, we did not show how rod rescue impacts RGC information rates at rod-mediated light levels. We were curious if there are similar differences in early, mid and late-treatment timepoints among RGCs under scotopic conditions as we have observed for photopic conditions. At a mean light level of 1 Rh*/rod/s, we presented a repeating checkerboard stimulus and calculated the mutual information between the stimulus and the RGC responses. In untreated animals, the rod-mediated information rate was undetectable due to the lack of CNG channels, which results in diminished photocurrent that leads to hyperpolarized rods[9]. However, early treatment restored rod-mediated signaling to WT levels by 7 M (3% lower than WT; Table 1, line 16) (Fig. 6C). Interestingly, the information rate at 4 M for early treatment was only 16.6 bits/s (17% lower than WT; Table 1, line 17), indicating it requires several months for information rates under scotopic conditions to reach normal levels following *Cngb1* rescue.

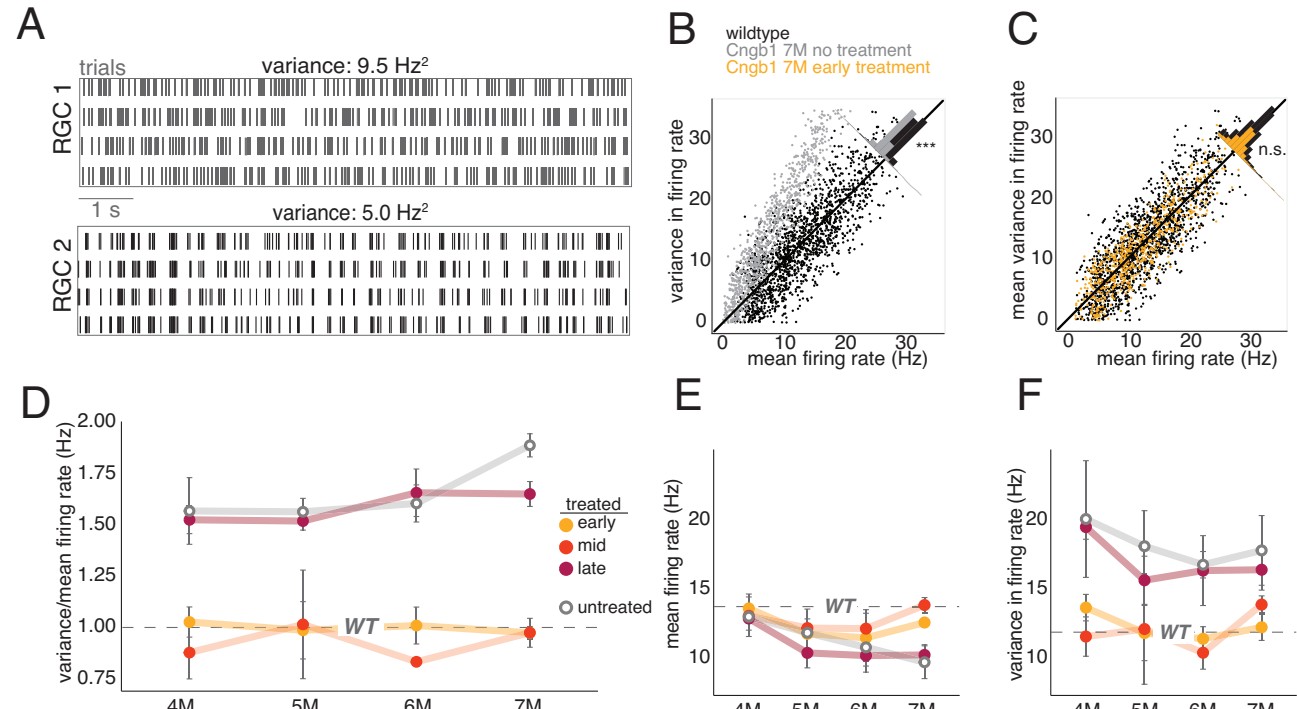

**Fig. 5 | Signal variance remains high following late treatment. A** Two example RGC responses to photopic checkerboard noise repeats with similar firing rates but different temporal variability in firing rates. **B** Variance and mean firing rates of RGCs in WT and 7 M untreated RGCs (temporal bin size was 5 ms). $P = 0.002$ **C** Same as (**B**), comparing WT and early treated RGCs, measured at 7 M post treatment.

$P = 0.32$. **D** Average variance/mean firing rate ratio +/−2 SD after treatment for different treatment timepoints. The dashed line shows WT value, which did not depend on age over the range of sampled timepoints. **E** Mean firing rate and **F** mean observed variance in firing rate +/−2 SD after treatment for different treatment timepoints. $n = 48$ retinas. Source data are provided as a Source Data file.

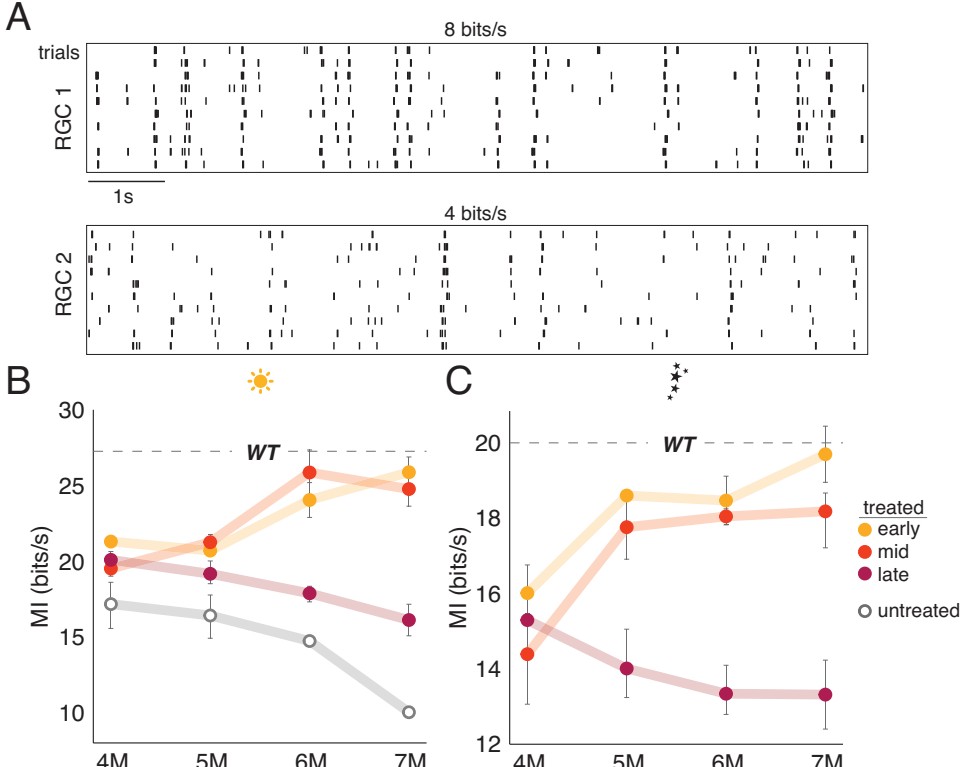

**Fig. 6 | Information transmission is not recovered following late treatment.**
**A** Two example RGC responses to photopic checkerboard noise repeats near the median information rate (top) and near the 25th percentile (bottom) in an experiment. Mean +/−2 SD mutual information rate of RGC responses (from 2 to 6 retinas) at **B** photopic and **C** scotopic light level from the top 10% most informative RGCs. Dashed line indicates the mean information rate observed in RGC responses from WT retinas. $n = 48$ retinas. Source data are provided as a Source Data file.

Similarly, mid-treatment also improved rod information transmission over several months, but the information rates did not quite recover to WT rates by 7 M (5 months post treatment). Late treatment also improved the information rate of the rod-mediated RGC signaling above baseline (which was essentially 0 bits/s in untreated animals), but the information rate declined over the months following treatment (Fig. 6C; 15% lower at 7 M than 4 M; Table 1, line 18), and never achieved information rates near those of WT RGCs (32% less than WT at 7 M; Table 1, line 19). Thus, late treatment was not sufficient to fully restore rod-mediated retinal signaling and signaling declined in the months following treatment. These results were also consistent for visual responses recorded at a mesopic light level (Supplementary Fig. S4). To ensure these results were not specific to a checkerboard stimulus, we also presented a repeated natural movie and obtained similar results under scotopic conditions (Supplementary Fig. S5).

## Discussion

In the coming years, significant resources will be devoted to the development of therapies for rare diseases, including those caused by *CNGB1* mutations (*CNGB1*-RP, a.k.a. RP45), which was recently highlighted as a key gene therapy target by the NIH-supported Accelerating Medicines Partnership Bespoke Gene Therapy Consortium. Defining gene therapy intervention points is essential to developing effective treatments that both restore and maintain vision. In this study, we addressed the question of how a preclinical mouse model for RP responds to a therapeutic intervention. We show that early (25% rod loss) and mid (50% rod loss) treatment did not completely halt photoreceptor loss. However, early and mid-treatment timepoints did restore rod- and cone-mediated visual signaling among RGCs. Interestingly, this restoration took a few months to complete, suggesting that some rewiring and/or circuit stabilization were required after

genetic rescue, though this needs to be investigated further to fully understand the recovery process. Following late intervention (70% rod loss), retinas exhibited evidence for persistent gliosis, suggesting persistent inflammation, and visual signaling among RGCs continued to steadily decrease under both rod- and cone-mediated conditions. These results indicate gene therapy interventions for *CNGB1*-RP may not succeed in preserving the remaining rod or cone vision if delivered after 50% rod loss.

Gene therapy is a growing field for a variety of photoreceptor degenerations, including RP (see review by Nuzbrokh et al.[34]). Several studies have assessed AAV-mediated gene replacement in both mouse and canine models of *CNGB1*-RP. Mouse studies found that regardless of intervention time or dose, *Cngb1* gene replacement improves function, but fails to halt cell death[35–37]. In addition, AAV-*Cngb1* treatment was compared between rodent and canine models of *CNGB1*-RP[38]. In both species, early intervention was correlated with improved results, indicating this trend is not mouse-specific. Considering these results from other studies, our findings indicate that insufficient vector dosage is not the primary factor limiting cell survival; rather, gene replacement is not enough to fully or immediately halt rod death. Importantly, we found that while some rod death continues after the early and mid-treatment timepoints (Fig. 1), retinal function (as assayed by RGC signaling) fully recovered following treatment (Figs. 4–6). In contrast, retinal function continued to deteriorate following the late treatment.

Studies on additional models of RP have shown greater viability in gene therapy success following late-stage intervention. Using an inducible cre system to correct *PDE6B*-RP, several studies investigated the differences between full rescue at early, mid, and late stages of degeneration, and partial rescue at an early timepoint[39–41]. The full rescue studies showed late intervention minimally improved scotopic

photoreceptor and bipolar cell function (as measured by electro-retinography, or ERG), but degeneration was halted at all treatment timepoints[39,41], though only one posttreatment timepoint was examined. The partial therapy failed to recover light responses at any dose and degeneration continued, though the authors attributed this to the death of untreated cells[40]. The cause of the discrepancy between our study showing continued cell death and these is unclear: differences could be caused by the mutation, intervention time, rate of degeneration, or timepoint(s) assayed post treatment. It is also possible that disease state, such as increased inflammation, plays a significant role in the amount of cell death occurring, at least in photoreceptor loss from *Cngb1* mutations.

A technical difference between our study and previous work on genetic rescue for RP is the use of large-scale MEAs to measure the physiological impact of genetic rescue rather than ERGs. ERGs have the advantage that they can be performed in vivo and can thus track the responses of photoreceptors and bipolar cells over time and following treatment in individual animals. They have the disadvantage that they average the responses of many cells and thus provide minimal insight into noise or signal fidelity, and mostly reveal changes in gain and response kinetics but without identifying the cellular location or molecular mechanism. Furthermore, they assay outer-retinal function, but photoreceptor degeneration could also alter inner-retinal function[11]. MEA measurements have the disadvantage that they are ex vivo and thus can only examine one timepoint per animal. However, MEA measurements have the potential advantages of providing more refined information about changes in receptive field structure, gain, noise, and information rates among individual and populations of RGCs, the output neurons of the retina. In the end, these methods are complementary approaches that can and should be used productively together to dissect the impact of retinal disease and genetic rescue across multiple aspects of retinal physiology[42,43].

People with RP are not typically diagnosed until after significant rod loss and the onset of cone dysfunction, particularly those with *CNGB1* mutations[44]. While the timeline of disease progression and diagnosis varies quite considerably, for *CNGB1*-RP median diagnosis occurs at 27 years, though most patients report nyctalopia at infancy/childhood and retain only central vision by their diagnosis. This timing is unfortunate based on the findings that early intervention results in better and more long-lasting visual function; by the time a patient is diagnosed, they are well past the early and mid-intervention windows of 25–50% rod loss that we examined. However, while late treatment does not halt degeneration and does not improve visual responses to normal, the improvements may be enough to restore behaviorally useful vision, and that vision may last for the lifetime of the patient. Future studies should assess visual behaviors in late-treated animal models.

Our study and others point to several additional therapeutic targets that may be important for improving outcomes for RP patients. First is targeting cone photoreceptors. Given that foveal cones persist much longer than rods and peripheral cones, preserving cone vision is an attractive target for new therapies. It is unknown why cones eventually die in this disease, despite not needing the causative gene (*CNGB1*) to function. There are several potential sources of continued degeneration that could be targeted in addition to gene replacement: metabolic stress, lack of rod-derived trophic factors, inflammation, or lack of structural support[45–47]. Thus, therapies aimed at cones have the benefit of having a long intervention timeframe, maintaining the range of vision that is most used by humans, and being mutation-independent, thus applicable to a wide variety of photoreceptor degenerations.

A second therapeutic target relates to reducing inflammation[17], which is a common effect of untreated retinal degeneration[17–20]. One sign of inflammation is activated glia, which were present in the late-treatment group of our study (Fig. 3), indicating the retina remained stressed despite curing the underlying cause of degeneration. While quantitative measurements were not performed, GFAP appeared more prevalent in late-treated than untreated retinas. Viral gene therapy will only worsen inflammation. Anti-inflammatories or other neuroprotective molecules may alleviate this form of disease pathogenesis, allowing the retina more opportunity to heal.

A third therapeutic target is synaptogenesis in the outer plexiform layer. While the retina appears to compensate for reduced synapses[4], it does need a certain number of intact photoreceptor-bipolar cell connections to maintain information flow. We found treated retinas had reduced outer plexiform structure relative to wild-type retinas, presumably from bipolar cells losing their presynaptic partners and downregulating their synaptic proteins. Increasing synaptogenesis is an attractive target to improve retinal signaling particularly for improving the outcomes of late treatment.

We found that light responses took several months to recover following gene correction, rather than several weeks. It remains unclear what circuit level changes occur during this dynamic period after therapy. Future studies are needed to yield insights into the mechanisms governing the recovery process, which could then be potentially harnessed to extend the window of therapy to later time points.

Finally, we recognize artificial gene replacement in a mouse model of disease is not translational to human therapeutics. Mouse retina lacks a cone-dense region and is (at best) more analogous to human peripheral retina[48]. Rather, these results yield insights into the best a therapy could achieve if improvements are made on the gene delivery front such as better cell penetration or expression of therapeutic genes. It is essential to perform these studies in additional models with retinas and visual systems which are more similar to human retina and human vision. For example, one key question that cannot be answered in rodent models is the impact of rod death on the macular region and fovea: does a higher density of cones lead to protection against the posttreatment degeneration seen in the *Cngb1*[neo/neo] mouse model? The development of non-human primate models is critical to answering this question and would lead to significant advances in treating retinal diseases.

## Methods

All research complies with relevant ethical regulations established by Duke University's Occupational Health and Safety Office and Institutional Animal Care and Use Committee.

### Animal model

Animal procedures were approved by the Duke University Institutional Animal Care and Use Committee guidelines (protocol A084-21-04) and in accordance with guidelines provided by the Association for Research in Vision and Ophthalmology. All experiments abide by ARRIVE guidelines. *Cngb1*[neo/neo] mice (sub-strain C57Bl/6J) were crossed with UBC-cre/ERT2[8] (JAX stock #007001, RRID: IMSR_JAX:007001) to generate mice with tamoxifen-inducible genetic rescue of *Cngb1*[3,9]. Mice were housed in a facility kept at approximately 72 °F and 30–70% humidity with a 12 h light/dark cycle and fed chow ad libitum. Both sexes were used (21 female, 19 male). Wild-type (*Cngb1*[+/neo] and *Cngb1*[+/+]) and untreated *Cngb1*[neo/neo] controls consisted of littermates; a portion of these animals were fed tamoxifen (only *Cngb1*[neo/neo] that were cre-). All genotyping was performed by Transnetyx using primers for neomycin FWD GGGCGCCCGGTTCTT, REV CCTCGTCCTGCAGT TCATTCA, PROBE ACCTGTCCGGTGCCC and WT allele FWD TCCT TAGGCTCTGCTGGAAGA, REV CAGAGGATGAACAAGAGACAGGAA, PROBE CTGAGCTGGGTAATGTC.

### Treatment

Tamoxifen chow (Envigo, TD.130858, 0.5 g/kg tamoxifen) replaced rodent chow (PicoLab 5053) for 7 days and was provided ad libitum.

Efficacy of tamoxifen treatment leading to genetic rescue has been previously described[9]. Mice were monitored during treatment to ensure no adverse effects to the drug. Mice that showed signs of illness were switched back to non-tamoxifen rodent chow and removed from the study. Additionally, select mice were confirmed to have recombined *Cngb1* using Transnetyx.

## Histology, microscopy, and quantification

Histology was performed on tissue from animals used in MEA experiments and was performed as previously described[4,9]. Briefly, enucleated eyes were fixed with 4% paraformaldehyde, hemisected, and back of the eye including retina submerged in 30% sucrose overnight at 4 °C. Eye cups were then submerged in Optical Cutting Temperature Media (Tissue-Tek, 4583), placed in a microcentrifuge tube and flash frozen. In all, 12-μm sections were cut using a Leica cryostat (CM3050). Slides were warmed to room temperature prior to staining, then rinsed three times with 1× phosphate-buffered saline (Santa Cruz, sc-296028), then incubated with 0.5% TritonX-100 (Sigma, X100) followed by 1% bovine serum albumin (VWR, 0332) for 1 h each. Primary antibodies were diluted with 0.3% TritonX-100 + 1% bovine serum albumin, applied to slides at 4 °C, and incubated overnight. Prior to applying secondary antibodies, slides were rinsed 3× with 1× phosphate-buffered saline. Secondary antibodies diluted with 1× PBS and incubated at room temperature for 1 h. Slides were then rinsed 3× with 1× phosphate-buffered saline, mounting media containing DAPI (Invitrogen, P36935) applied, coverslipped, and sealed with clear nail polish. Antibodies used included mCar (1:500, Millipore AB15282, RRID: 652 AB_1163387), PCP2 (1:500, Santa Cruz sc-137064, RRID: AB_2158439), CtBP2 (1:1000, BD Biosciences Cat# 612044, RRID:AB_399431), GFAP (1:400, Sigma-Aldrich Cat# G3893, RRID:AB_477010), CNGA1 (1:50, generously provided by R. Molday)[49], Alexa Fluor donkey anti mouse 647 (1:500, Thermo Fisher Scientific Cat# A-31571, RRID:AB_162542), and Alexa Fluor donkey anti rabbit 555 (Thermo Fisher Scientific Cat# A-31572, RRID:AB_162543).

All images were taken from dorsal retina unless otherwise noted. Light microscope images were captured with a Zeiss Axioplan2 microscope using a 63× air objective. Confocal images were taken using a Nikon AR1 confocal microscope using a 60× oil objective and motorized stage. Images were processed using FIJI software[50].

Photoreceptor quantification was performed across three steps, which included preprocessing, automated quantification, and then manual count corrections. Images were processed to include 15 z-slices (0.5 μm each) of the outer nuclear layer. Image contrast was then adjusted to maximize local contrast using the Integral Image Filters plugin at default parameters. Next, we used a custom Matlab script to detect nuclei in the preprocessed images. For each image, we sampled three rectangular regions (1000 square microns) that were located approximately center, left, and right within the ONL. These three counts were averaged for each retina. Images were converted to a binary map to detect high-intensity areas using a threshold of 0.4 or a pixel value of 104 for 8-bit images. These high-intensity areas were restricted to those that had a minimum width of 2.1 μm and were located at least 1.05 μm away from its neighbors. Due to these parameters, approximately 5–10% of visible nuclei or ~5–15 nuclei in each sampled region were out of focus and were not automatically counted. Thus, in the final step, nuclei which were missed were manually counted.

## MEA recordings

All recordings were performed as described previously[4]. Briefly, mice were dark-adapted overnight by placing their home cage with food and water into a light-shielded box fitted with an air pump. All procedures were performed under IR illumination using night vision goggles. Mice were euthanized via decapitation. Following decapitation, eyes were enucleated and placed in bubbled room temperature Ames media (Sigma, A1420) for the duration of the dissection. Eyes were then hemisected, vitrectomy performed, and retina detached from RPE and sclera. A ~1 × 2 mm dorsal retinal piece was cut and placed RGC side down on a 519 dense multielectrode array with 30-μm spacing[25,51,52]. Throughout the recording, 32 °C bubbled Ames was refreshed at a rate of ~6 mL/min.

## Spike sorting and neuron identification

Spikes for each of the electrodes were identified by using four times the voltage standard deviation[53,54]. Spike sorting was performed by an automated PCA algorithm described previously[26]. To track identified RGCs across light conditions and stimuli, cell clusters were sorted in the same PCA subspace at each light level. Neurons were verified as matches across stimulus and light conditions by examining their spike waveforms and electrical images[26]. RGCs were classified at the photopic light level by clustering cells according to their receptive field properties and spike-train autocorrelation functions.

## Visual stimuli

Visual stimuli were described previously[4]. Visual stimuli consisted of binary checkerboard noise and natural movies[55,56], presented using a gamma-calibrated OLED display (Emagin, SVGA + XL Rev3) focused by a 4× objective (Nikon, CFI Super Fluor ×4) attached to an inverted microscope (Nikon, Ti-E). For scotopic (~1 Rh*/rod/s) and mesopic (~100 Rh*/rod/s) checkerboard stimuli, each square was ~150 × 150 μm and refreshed every 66 ms. Photopic (~10,000 Rh*/rod/s) checkerboard squares were each 75 × 75 μm and refreshed every 33 ms. Repeat movies consisted of a 10 s clip of either checkboard or natural movies repeated 100×. Checkerboard repeats had the same parameters at a given light level as described above.

## Spike-triggered averaging and nonlinearity calculations

Spike-triggered averaging (STA) was used to estimate the linear component of the receptive fields for each RGC. Procedures for calculating the spatiotemporal receptive field were identical to those described previously[4]. We analyzed the spatial and temporal receptive fields of RGCs for which at least 60% of the variance in the STA was captured by a rank-one factorization. Cells which had a temporal filter that were biphasic were included because a well-defined zero crossing time was necessary for estimating the time-to-zero. 85% of space-time separable STAs met this criterion. Static nonlinearities were calculated for RGCs with space-time separable STAs by mapping the convolution of the linear filter and checkerboard stimulus with their response[27]. These static nonlinearities were used to characterize the contrast response function of individual RGCs and their response gain.

## Mutual information

Shannon's definition of entropy was used to measure mutual information with the use of the direct method described previously[4,57]. Spike trains were binned according to entropy estimates that achieved the Ma Upper Bound[58] and ranged from bins of 4–6 ms and formed patterns of 3–6 bins. Mutual information was thus computed as:

$$I(S;R) = H(R) - H(R;S) \tag{1}$$

where $H(R)$ is the entropy in the response and $H(R;S)$ is the entropy of the response conditioned on the stimulus.

## Statistical analysis

Two-sided Kolmogorov–Smirnov tests were used to determine differences between treatment groups. *P* values were corrected for multiple comparisons by Bonferroni correction. To measure whether differences across timepoints could be produced by other factors (e.g., experiment-to-experiment variability), a parametric linear mixed-effects model was used[59]. The mixed-effects model accounts for

retina-to-retina variability by adding each experiment as a random effect. This procedure permitted making broad-level inferences about the RGC populations without dependence on experimental variability. In addition, the sex of the animal and a neuron's cell type were considered by including them as interaction terms with the treatment conditions. This step enabled determining whether treatment conditions were associated with information rates and receptive field sizes in a sex-independent fashion. The model indicated that conclusions about the impact of genetic rescue on RGC signaling were insensitive to sex, experimental variability, and cell type.

### Reporting summary
Further information on research design is available in the Nature Portfolio Reporting Summary linked to this article.

## Data availability
The microscopy data generated in this study have been deposited in the Dryad database under the accession code https://doi.org/10.5061/dryad.rv15dv4bv. The MEA data are available under restricted access due to the size of the unprocessed files. Data will be shared for any non-commercial purpose. Access can be coordinated by contacting the corresponding author Greg Field (GField@mednet.ucla.edu), who will respond to requests within 5 business days. The processed MEA data are available at the github repository https://github.com/mishek-thapa/treatment_paper and also provided in the Source Data file, provided with this manuscript. Source data are provided with this paper.

## Code availability
Source code used to generate figures are available in the GitHub repository https://github.com/mishek-thapa/treatment_paper. The code used to calculate mutual information can be found in the GitHub repository https://github.com/mishek-thapa/cng-data[4].

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

## Acknowledgements

We thank our funding sources: National Institute of Health (NIH) R01 EY027193 (G.D.F., A.P.S., and J.C.), NIH NEI core grant EY5722 to Duke University, Holland Trice Foundation (G.D.F. and M.L.S.), Whitehead Foundation (G.D.F.), and Research to Prevent Blindness unrestricted grant to Duke University. We thank Dr. S. Roy for helpful discussions on information theory.

## Author contributions

Experimental design: M.L.S., J.C., A.P.S., and G.F. Histology experiments: M.L.S. and T.W. MEA experiments: M.L.S. Histology analysis: M.L.S., M.T., and J.C. MEA analysis: M.L.S., M.T., and G.D.F. Writing: M.L.S., M.T., and G.D.F. Editing: M.L.S., M.T., and G.D.F.

## Competing interests

The authors declare no competing interests.
