## [Peer Review File · Nature Communications]

REVIEWER COMMENTS

Reviewer #1 (Remarks to the Author):

The authors present a manuscript describing that the ability to anatomically (histology) and functionally (MEA) rescue the phenotype of a Cngb1 mouse model of rod-cone dystrophy decreases with advancing rod loss. Specifically, the retina is more rescuable when intervention occurs at an early- or mid- time point when compared to intervention at a late-time point. Conceptually, this is not a particularly novel observation, where disease stage (e.g. number of cells remaining, retinal lamination etc) has long been considered an essential factor when designing a therapeutic intervention in either an artificial model like a mouse, or in human clinical trials, where patient selection and outcome measures are critical to success. However, this study would provide valuable empirical data describing the importance of correctly timing an intervention that would be of interest to the vision science community. The study is very well carried out with appropriate considerations to experimental details, controls, statistics and provides a mostly comprehensive discussion with only a few general areas for improvement of clarity.

Comments

- The authors state that there is 25%, 50% and 75% rod loss at 1M, 2M and 3M, respectively, with cone loss of ~5% at 3M only, citing Scalabrino et al 2022 (eLife) in support. The authors do not take into account that the model is actually quite variable, especially with regards to cone loss, and how this might impact the degree of rescue. The authors should include actual percentages of photoreceptor survival measured during this study with plus-minus values to show variability.
- It is stated that 'sparser synapses following late treatment suggest that earlier treatment may be important for rescuing normal and long-lasting retinal function' (Lines 106 – 108), but it is unclear whether the observed reduction in synapses is due to there being fewer total cells remaining to form synaptic connections, or a down regulation of CtBP2 at the termini? This needs clarifying.
- GFAP appears to be more prevalent in late treated mice than in untreated animals. The authors should expand on the reason for this observation and whether it potentially indicates a pro-inflammatory response to 'treatment'. If so, why would this not be observed at early- and mid- timepoints.
- The authors mention in the discussion that the slow recovery (e.g. of cone-mediated visual signalling) following early- or mid- treatment towards wild type maybe due to 'rewiring/circuit stabilization'. Is there any empirical evidence for rewiring taking place over a several month time scale to support this supposition?
- The authors do not go into sufficient detail in the discussion relating to how treatment of rods in a mouse – where rods and cones are relatively evenly distributed – would compare to a human scenario, where cones are centralized to the fovea with total rod exclusion. Given these anatomical differences, would treatment of peri-foveal rods be sufficient to preserve foveal cones?

Reviewer #2 (Remarks to the Author):

This is an excellent manuscript and I very much enjoyed reading it. The authors describe a mouse model of retinitis pigmentosa that can be rescued without gene therapy. They employ animals with a disrupted CNGB1 gene with a flexed Neo cassette in intron 19 and crossed onto a background with an tamoxifen inducible cue-recombinase. This allows them to rescue the phenotype at different time courses of the disease and examine how disease progression and function are affected once CNGB1 expression is restored.

They find that rescue when 25% or 50% of rods cells are lost restores function similar to that of wild-type levels, but does not fully stop progression. Interestingly, rescue at a later time point when on 70% of rod cells are lost has a worse outcome and also demonstrate irreversible gliosis. I think the experiments are well designed and the data are quite convincing. The ganglion cell recordings are quite elegant and demonstrate how this technique can be informative and complimentary to ERG. The paper is quite important in that it suggests that gene therapy will likely be more effective if delivered early and there maybe a point of no return after which the benefits of gene therapy might diminish considerably.

I only have several minor comments and suggestions to improve the paper

line 15 - I think it is important emphasize that most RP patients never go completely blind (less than 1%), however the majority do reach "legal" blindness

line 23 - In general I think the authors need to be conservative in translating their conclusion from rodents to how gene therapies might work in humans. The rodent retina is perhaps analogous to the human midperipheral retina. Human foveal cones may behave quite different from cones locate further peripherally and it is the case that foveal cones often survive longer and for many years after rods have disappeared. I think it is fair to say that the timing of gene therapy may affect the potential to rescue peripheral vision, but the implications for the impact on central vision are less clear.

Line 46 - Can you authors give an estimation of the efficiency of restoration of CNGB3 levels following tamoxifen treatment? Have you done western blot to compare to wild-type levels? Does it work in all cells and could the progression that is seen be a result of cells where excision failed?

The second to last paragraph about cricket hunting, etc.. is interesting but could probably be eliminated.

Reviewer 1

The authors present a manuscript describing that the ability to anatomically (histology) and functionally (MEA) rescue the phenotype of a Cngb1 mouse model of rod-cone dystrophy decreases with advancing rod loss. Specifically, the retina is more rescuable when intervention occurs at an early- or mid- time point when compared to intervention at a late-time point. Conceptually, this is not a particularly novel observation, where disease stage (e.g. number of cells remaining, retinal lamination etc) has long been considered an essential factor when designing a therapeutic intervention in either an artificial model like a mouse, or in human clinical trials, where patient selection and outcome measures are critical to success. However, this study would provide valuable empirical data describing the importance of correctly timing an intervention that would be of interest to the vision science community. The study is very well carried out with appropriate considerations to experimental details, controls, statistics and provides a mostly comprehensive discussion with only a few general areas for improvement of clarity.

Comments

- The authors state that there is 25%, 50% and 75% rod loss at 1M, 2M and 3M, respectively, with cone loss of ~5% at 3M only, citing Scalabrino et al 2022 (eLife) in support. The authors do not take into account that the model is actually quite variable, especially with regards to cone loss, and how this might impact the degree of rescue. The authors should include actual percentages of photoreceptor survival measured during this study with plus-minus values to show variability.

We thank the reviewer for raising this point. Individual data points are available in the Data Availability section. We have added plus-minus values to lines 89-93. We have also added text to Line 53 “We treated mice at 1, 2, and 3 months of age corresponding to **approximately** 25%, 50% and 70% rod loss, and 0%, 0% and 5% cone loss, respectively.”

- It is stated that ‘sparser synapses following late treatment suggest that earlier treatment may be important for rescuing normal and long-lasting retinal function’ (Lines 106 – 108), but it is unclear whether the observed reduction in synapses is due to there being fewer total cells remaining to form synaptic connections, or a down regulation of CtBP2 at the termini? This needs clarifying.

We agree with the reviewer that the cause of reduced synapses is unclear and likely a combination of fewer synapses + downregulation of synaptic proteins. We added “(See discussion)” to line 109. We also added to Lines 328-329 “We found treated retinas had reduced outer plexiform structure relative to wild-type retinas, presumably from bipolar cells losing their pre-synaptic partners **and downregulating synaptic proteins.**”

- GFAP appears to be more prevalent in late treated mice than in untreated animals. The authors should expand on the reason for this observation and whether it potentially indicates a pro-inflammatory response to ‘treatment’. If so, why would this not be observed at early- and mid- timepoints.

We thank the reviewer for this comment and agree GFAP appears more prevalent in late treated mice than in untreated animals. We verified this in additional retinas and sections to ensure it isn't image selection bias.

We have clarified the inflammation is a consequence of the disease and not from the treatment at lines 117-118 **"as well as in tamoxifen treated WT retinas."** and lines 317-318 **"Inflammation is a common effect of untreated retinal degeneration"**¹⁷⁻²⁰

We also added lines 320-322 to Discussion **"While quantitative measurements were not performed, GFAP appeared more prevalent in late treated retinas than in untreated, though the reason for this is unclear as the tamoxifen treatment alone did not induce observable GFAP labeling in WT retinas (data not shown)."**

- The authors mention in the discussion that the slow recovery (e.g. of cone-mediated visual signalling) following early- or mid- treatment towards wild type maybe due to 'rewiring/circuit stabilization'. Is there any empirical evidence for rewiring taking place over a several month time scale to support this supposition?

To our knowledge, there is no empirical evidence of this. This is why the statement is in the Discussion and we say this is a possibility (it is suggested by the data) – there may be other explanations too. Studies related to homeostatic plasticity show single timepoints (for example Johnson 2017, Care 2020). Koch et al 2017 did artificial rescue with Pde6b mutants but did not track function at multiple timepoints.

In general, the timelines in viral gene therapy vary (for example Jacobson et al 2015 shows in humans treated with Luxterna improved over 1-3 years before visual decline, whereas Wagner et al 2021 shows improvement at 3M followed by decline). These results could be attributed to continued cell loss rather than circuit changes and are global retinal measurements rather than single cell population measurements like we performed.

To further soften and clarify the statement, we have made the following edit: Lines 252-253 **"Interestingly, this restoration took a few months to complete, suggesting that some rewiring and/or circuit stabilization were required after genetic rescue, though this needs to be investigated further to fully understand the recovery process."**

- The authors do not go into sufficient detail in the discussion relating to how treatment of rods in a mouse – where rods and cones are relatively evenly distributed – would compare to a human scenario, where cones are centralized to the fovea with total rod exclusion. Given these anatomical differences, would treatment of peri-foveal rods be sufficient to preserve foveal cones?

We thank the reviewer for bringing up this interesting point. We agree this is an important question. As the reviewer appreciates, it is difficult to generalize from results in a mouse, which lacks a fovea, to primates. To be blunt, we don't know what will happen in primates when rods near the fovea are rescued and how this will impact the function or survival of foveal cones. Rather than speculate, we thought it would be useful to raise this as a question and promote the need for primate models of retinal

degeneration to address these critical questions in the retinal degeneration and gene therapy fields. We have modified the end of the Discussion to include the need for primate models of retinal degeneration to answer these questions (lines 338-339; 342-346).

Reviewer 2

This is an excellent manuscript and I very much enjoyed reading it. The authors describe a mouse model of retinitis pigmentosa that can be rescued without gene therapy. They employ animals with a disrupted CNGB1 gene with a flexed Neo cassette in intron 19 and crossed onto a background with an tamoxifen inducible cue-recombinase. This allows them to rescue the phenotype at different time courses of the disease and examine how disease progression and function are affected once CNGB1 expression is restored.

They find that rescue when 25% or 50% of rods cells are lost restores function similar to that of wild-type levels, but does not fully stop progression. Interestingly, rescue at a later time point when on 70% of rod cells are lost has a worse outcome and also demonstrate irreversible gliosis. I think the experiments are well designed and the data are quite convincing. The ganglion cell recordings are quite elegant and demonstrate how this technique can be informative and complimentary to ERG. The paper is quite important in that it suggests that gene therapy will likely be more effective if delivered early and there maybe a point of no return after which the benefits of gene therapy might diminish considerably.

I only have several minor comments and suggestions to improve the paper

line 15 - I think it is important to emphasize that most RP patients never go completely blind (less than 1%), however the majority do reach "legal" blindness.

We thank the reviewer for this important point and have added the sentence **“This cell loss greatly diminishes vision, with most patients becoming legally blind.”** to line 15.

line 23 - In general I think the authors need to be conservative in translating their conclusion from rodents to how gene therapies might work in humans. The rodent retina is perhaps analogous to the human midperipheral retina. Human foveal cones may behave quite different from cones locate further peripherally and it is the case that foveal cones often survive longer and for many years after rods have disappeared. I think it is fair to say that the timing of gene therapy may affect the potential to rescue peripheral vision, but the implications for the impact on central vision are less clear.

We agree. Reviewer 1 also recommended some discussion regarding mouse versus human. We have added lines 338-339 and 342-346 to indicate the importance of being circumspect about generalizing results in the mouse retina to human, and the need for

primate models of retinal degeneration to understand how foveal cones are impacted by rod degeneration and rod rescue.

“Finally, we recognize artificial gene replacement in a mouse model of disease is not translational to human therapeutics. **Mouse retina lacks a cone dense region and is (at best) more analogous to human peripheral retina**⁴⁸. Rather, these results yield insights into the best a therapy could achieve if improvements are made on the gene delivery front such as better cell penetration or expression of therapeutic genes. It is essential to perform these studies in additional models with retinas and visual systems which are more similar to human retina and human vision. **For example, one key question that cannot be answered in rodent models is the impact of rod death on the macular region and fovea: does a higher density of cones lead to protection against the continued degeneration seen in the Cngb1^{neof/neo} mouse model? Development of non-human primate models are a critical step to answering this and related questions, and it would lead to significant advances in understanding retinal disease and therapy.**”

Line 46 - Can you authors give an estimation of the efficiency of restoration of CNGB3 levels following tamoxifen treatment? Have you done western blot to compare to wild-type levels? Does it work in all cells and could the progression that is seen be a result of cells where excision failed?

We assume the reviewer meant “Cngb1” not “Cngb3”. The efficacy of tamoxifen induced recombination was published in Wang et al., 2019. Mutant transcripts are not detectable in retinas after 7 days of tamoxifen treatment, though deep sequencing has not been performed, which would allow us to confidently quantify the efficiency of restoration. Protein levels in treated mice are similar to WT levels (Figure 3, Wang et al., 2019). We added in a reference to this work in Lines 79-80 “**(prior analysis of recombination provided in Wang et al., 2019 and prior quantification of cone loss provided in Scalabrino et al., 2022).**”

REVIEWERS' COMMENTS

Reviewer #1 (Remarks to the Author):

The authors have addressed my concerns.

Reviewer #2 (Remarks to the Author):

The authors have addressed all my concerns.